# A β-Hairpin Motif in the Envelope Protein E2 Mediates Receptor Binding of Bovine Viral Diarrhea Virus

**DOI:** 10.3390/v13061157

**Published:** 2021-06-17

**Authors:** Fernando Merwaiss, María José Pascual, María Trinidad Pomilio, María Gabriela Lopez, Oscar A. Taboga, Diego E. Alvarez

**Affiliations:** 1Instituto de Investigaciones Biotecnológicas (IIB), Universidad Nacional de San Martín (UNSAM), Consejo Nacional de Investigaciones Científicas y Técnicas (CONICET), Buenos Aires B1650HMR, Argentina; fmerwaiss@iibintech.com.ar (F.M.); mjpascual@iib.unsam.edu.ar (M.J.P.); trinidadpomilio@gmail.com (M.T.P.); 2Instituto de Agrobiotecnología y Biología Molecular (IABIMO), Instituto Nacional de Tecnología Agropecuaria (INTA), CONICET, Buenos Aires B1686IGC, Argentina; lopez.mariag@inta.gob.ar (M.G.L.); taboga.oscaralberto@inta.gob.ar (O.A.T.)

**Keywords:** pestivirus, entry, virus-receptor interaction

## Abstract

Pestivirus envelope protein E2 is crucial to virus infection and accomplishes virus-receptor interaction during entry. However, mapping of E2 residues mediating these interactions has remained unexplored. In this study, to investigate the structure-function relationship for a β-hairpin motif exposed to the solvent in the crystal structure of bovine viral diarrhea virus (BVDV) E2, we designed two amino acidic substitutions that result in a change of electrostatic potential. First, using wild type and mutant E2 expressed as soluble recombinant proteins, we found that the mutant protein had reduced binding to susceptible cells compared to wild type and diminished ability to inhibit BVDV infection, suggesting a lower affinity for BVDV receptors. We then analyzed the effect of β-hairpin mutations in the context of recombinant viral particles. Mutant viruses recovered from cell culture supernatant after transfection of recombinant RNA had almost completely inhibited ability to re-infect susceptible cells, indicating an impact of mutations on BVDV infectivity. Finally, sequential passaging of the mutant virus resulted in the selection of a viral population in which β-hairpin mutations reverted to the wild type sequence to restore infectivity. Taken together, our results show that this conserved region of the E2 protein is critical for the interaction with host cell receptors.

## 1. Introduction

Bovine viral diarrhea virus (BVDV), the prototype member of the pestivirus genus, is a positive-strand RNA virus that infects cattle and causes major economic losses to the livestock industry worldwide. The genus Pestivirus (family *Flaviviridae*) also includes pathogens of domestic animals such as pigs (classical swine fever virus, CSFV) and sheep (border disease virus, BDV) and other viruses infecting wild ruminants and boars. Based on the broad host range of pestiviruses, a recent revision of taxonomy of the genus has proposed naming species in a host-independent manner [1]. Thus, according to sequence analysis pestiviruses are classified into Pestivirus A (original designation bovine viral diarrhea virus 1), Pestivirus B (bovine viral diarrhea virus 2), Pestivirus C (classical swine fever virus), Pestivirus D (border disease virus, BDV), Pestivirus E (pronghorn pestivirus), Pestivirus F (Bungowannah virus, BuPV), Pestivirus G (giraffe pestivirus), Pestivirus H (Hobi-like pestivirus), Pestivirus I (Aydin-like pestivirus), Pestivirus J (rat pestivirus) and Pestivirus K (atypical porcine pestivirus, APPV).

The RNA genome of Pestiviruses is translated into a single polyprotein that is cleaved by viral and cellular proteases to yield structural and non-structural proteins, namely, N^pro^-C-E^rns^-E1-E2-p7-NS2-NS3-NS4A-NS4B-NS5A-NS5B. Structural glycoproteins E^rns^, E1 and E2 are associated to the lipid envelope of the virus that surrounds the nucleocapsid comprised of the capsid protein C and the virus genome. Cleavage for the release of NS3 at the NS2-3 junction distinguishes cytopathic and non-cytopathic pestivirus biotypes and is achieved by the insertion of viral and host protease target sequences upstream of NS3. While non-cytopathic biotypes circulate in the field and can establish persistent infection upon crossing the placenta of the dam to infect the fetus, cp viruses typically arise in persistently infected animals as a result of a recombination event in the infecting ncp virus and lead to fatal mucosal disease [2].

E2 is a 53–55 kDa glycoprotein that is responsible for the interaction of Pestiviruses with cellular receptors, it determines the cellular tropism [3], and it is a main antigenic determinant of infection and thus the target of neutralizing antibodies against this group of viruses [4,5,6,7,8,9]. Consistent with its role as a receptor binding protein, soluble E2 blocks viral infection [10]. Furthermore, blocking is effective with either CSFV or BVDV E2 on porcine and bovine cells, suggesting usage of the same receptor by both viruses [11]. To date, bovine CD46 is the only identified BVDV receptor and monoclonal antibodies against CD46 block virus infection [10,12,13]. Interestingly, a recent study based on the construction of a knock-out cell line showed that different BVDV isolates display markedly reduced infection of CD46 deficient cells, but viruses that escape CD46 dependency were rescued upon passaging in knock-out cells [14]. In spite of increasing virus attachment in heterologous cell lines, overexpression of bovine CD46 is not sufficient to turn cells susceptible to infection [13]. In turn, CSFV is still able to enter different cell lines deficient of porcine CD46 [15]. Thus, the use of co-receptors appears as a common theme in pestivirus entry. 

The crystal structure for BVDV E2 has been resolved, and it revealed a three-domain architecture with two IgG-like distal domains (domains I and II) and a membrane proximal domain with a unique elongated fold (domain III) [16,17]. Domain II exposes to the solvent a β-hairpin motif encompassing a 12-amino acid stretch. The motif comprises two β-sheets connected by a projecting loop. Substitution analysis of CSFV by BVDV sequence within this motif indicated that this stretch is critical for replication of CSFV in porcine cells [18,19]. In addition, CSFV competes with a peptide mapping to this motif for binding to virus susceptible cells [20]. Finally, this stretch of amino acids represents a conserved linear epitope among CSFV and characterization of neutralization escape variants identified specific amino acid substitutions in this region [21]. Based on these lines of evidence, it has been proposed that the β-hairpin is involved in virus-receptor interaction for CSFV. However, direct evidence on the function of this motif in the receptor binding capacity of BVDV is still lacking.

Here, we addressed a structure-function analysis of the exposed β-hairpin using BVDV as a model. By combining functional analysis of wild type and mutated versions of soluble E2 with the assessment of the impact of mutations in the β-hairpin in the context of recombinant BVDV genomes, we pinpointed residues that are critical for receptor binding and virus infectivity.

## 2. Materials and Methods

### 2.1. Cells and Viruses 

MDBK (Bos taurus kidney, ATCC CCL-22) cells were grown in Dulbecco’s modified Eagle medium (DMEM) supplemented with 10% fetal bovine serum and antibiotics under 5% CO⁠_2_ at 37 °C. For infections, cells were cultivated in DMEM supplemented with 2% Horse serum and antibiotics under 5% CO⁠_2_ at 37 °C. Spodoptera frugiperda Sf9 cells were grown in EX-Cell 420 serum-free medium (Sigma, St. Louis, MO, USA) supplemented with antibiotic-antimycotic (Gibco) at 28 °C.

Infectious viral particles of the cp- and ncp-BVDV strain NADL were obtained from the pACNR/NADL and pACNR/cIns^-^ NADL infectious cDNA clones, respectively (kindly provided by Charles Rice, The Rockefeller University), by reverse genetics [22]. Briefly, cDNA infectious clones were linearized at the SbfI restriction site and were used as templates for in vitro transcription by use of T7 RNApol (Ambion, Austin, TX, USA) according to the manufacturer’s instructions. Synthetic RNAs were treated with DNase I and transfected into MDBK cells by use of Lipofectamine 2000 (Invitrogen, Waltham, MA, USA). Infectious BVDV particles were recovered from the cell culture supernatant and stored at −70 °C until use.

### 2.2. Virus Titration by qPCR

RNA was extracted from cell culture supernatant with TRIzol (Invitrogen) according to the manufacturer instructions and used as a template for reverse transcription with Moloney murine leukemia virus (MMLV) reverse transcriptase (Promega) and random hexamers. To quantify BVDV genomic RNA, a 108-bp fragment from the 5′ non-translated region (NTR) was amplified with primers 5′NTR forward (5′-GAGGCTAGCCATGCCCTTAGT-3′) and 5′NTR reverse (5′-TCGAACCACTGACGACTACCCT-3′). The reaction was carried out in 2× SYBR green PCR master mix (Applied Biosystems) in a Step One Plus apparatus (Applied Biosystems), using an experimental run protocol of 10 min of activation at 95 °C, followed by 45 cycles of amplification and quantification (15 s at 95.0 °C, 1 min at 60.0 °C, and 35 s at 78.5 °C), when SYBR green I signal was measured. Assays were conducted in triplicates, and a standard curve was constructed using 10-fold serial dilutions of pACNR/NADL DNA plasmid to estimate the number of gRNA copies/mL.

### 2.3. Construction of Recombinant ncpBVDV Carrying Mutations in E2

An overlapping PCR product with mutations coding for N144A and T147A amino acid changes was generated with primers 24 (5′-GAGGGAAGTTCAATACAACG-3′) and 74 (5′-GTTGTGGCTAAGGCGTCCATCGCGAATGACGTAC-3′), and primers 73 (5′-GTACGTCATTCGCGATGGACGCCTTAGCCACAAC-3′) and 39 (5′-TTCCTGCCTGAAGGGCCCCTCAAAGTTGCCTTC-3′). The amplicon was digested with RsrII and BglII restriction enzymes and the resulting fragment subcloned into pACNR/cIns^-^ NADL.

### 2.4. Recombinant E2 Expression and Purification 

Wild-type and mutant recombinant E2 were produced using the baculovirus system as it was previously described by our group [10]. Briefly, the E2 sequence amplified from the pACNR/NADL infectious cDNA clone without its transmembrane domain and carrying a C-terminal 6xHis tag was cloned between SmaI and PstI restriction sites downstream of the baculovirus GP64 signal peptide sequence in the transfer vector pFBSD (derived from pFastBac1 [Invitrogen]). For this work, we incorporated between SmaI and RsrII restriction sites a synthetic sequence (Genescript) containing the Biotin Acceptor Peptide (BAP, GLNDIFEAQKIEWHE) coding sequence and a unique XbaI site upstream of the first 372 nucleotides (nt) of E2. Thus, we generated the transfer vector pFBSD/ps-BAP-E2Δ-6xHis, which codes for a soluble and secreted version of E2 flanked by the BAP and 6xHis epitopes. The mutant version of this plasmid incorporating amino acid changes N144A and T147A was generated by restriction-free cloning [23] using primers 73 and 74. Transfer vectors were transformed into Escherichia coli DH10BAC cells (Invitrogen) to generate a recombinant bacmid DNA carrying the E2 gene expression cassette. For recombinant baculovirus generation, 7 × 10^5^ Sf9 cells were seeded into 6-well plates and transfected with bacmid DNA by use of Lipofectamine 3000 reagent (Invitrogen) according to the manufacturer’s protocol. After 5 days of incubation at 28 °C, the supernatant was harvested and clarified. For viral titer amplification, successive rounds of infection of Sf9 cells were performed. Virus titers were determined by an endpoint dilution assay by using a GFP infection-induced cell line [24].

In order to obtain E2 directly biotinylated from cell supernatant, an Sf9 cell line was generated and selected expressing the *E. coli* BirA protein. For this, the BirA sequence of the pLenti4sBirA vector [25] (a gift from Brett Lindenbach, Yale University) was subcloned between BamHI and XhoI sites from the insect cells expression plasmid pIB (Invitrogen). In this construct, a signal sequence (amino acids 756 to 778 of yellow fever virus strain 17D; GenBank accession no. X03700) incorporated upstream of BirA directs BirA to the secretory compartment. Recombinant proteins were detected by Western blotting, employing both anti-6xHis epitope tag antibody (Rockland) and Streptavidine-HRP. For final protein production, five T175 flasks were seeded with 2 × 10^7^ Sf9 cells each and infected at a multiplicity of infection (moi) of 5. E2 purification was performed with a Ni-Sepharose high-performance column (GE Healthcare) according to the manufacturer’s instructions. After loading, protein fractions were eluted in a step gradient of imidazole and analyzed by SDS-PAGE and Coomassie blue staining.

### 2.5. Fluorescence Microscopy

MDBK cells were seeded onto glass coverslips in 24 well plates at a density of 10^5^ cells/well, allowed to attach overnight and transfected with 1 µg of in vitro transcribed RNA or infected at a moi of 1. At the indicated time points, cells were thoroughly washed and fixed using paraformaldehyde (PFA) 4%. Fixed samples were first incubated with a mouse polyclonal antibody against BVDV E2 [10] and then washed three times in phosphate-buffered saline (PBS) before addition of an Alexa Fluor-conjugated secondary antibody. Cell nuclei were stained with DAPI (4,6-diamidino-2-phenylindole), and coverslips were then mounted onto glass slides by use of FluoroGuard antifade reagent (Bio-Rad, Hercules, CA, USA). Samples were visualized under a Nikon Eclipse 80i fluorescence microscope equipped with a DS-Qi1Mc camera and images processed with ImageJ software.

### 2.6. Binding Assays

MDBK cells were seeded onto 24 well plates and allowed to attach overnight. Cells were incubated with either WT or mutant recombinant E2 at different concentrations for one hour at room temperature. Detection of attached E2 was performed using both flow cytometry and Western blot as further described.

#### 2.6.1. E2-Binding Detected by Western Blot 

Cells were thoroughly washed and lysed in cracking buffer (2% SDS, 10% glycerol, 60 mM Tris-HCl, pH 6.8, 0.1 M dithiothreitol [DTT]). Samples were resolved by SDS-PAGE. E2 was detected by Western blotting employing a mouse polyclonal antibody against E2, and actin detection with a polyclonal antibody against actin was used a loading control. 

#### 2.6.2. E2-Binding Detected by Flow Cytometry 

Cells were thoroughly washed, lifted using an EDTA-PBS solution and fixed with PFA 4%. Samples were stained using a polyclonal antibody against E2 produced in mouse [10] and a secondary antibody conjugated to Alexa Fluor 488. The fluorescence signal was measured using a flow cytometer (CyFlow^®^ Space, Partec, Germany) at a detection spectrum of 488 nm. Data were analyzed in the FlowJo 7.6.2 software package. 

### 2.7. Cytopathic Effect Reduction Assay

Cytopathic effect reduction assays were carried out as previously described [26]. Briefly, confluent monolayers of MDBK cells in 96 well plates (approximately 15,000 cells per well) were infected with cpBVDV at a multiplicity of infection (moi) of 0.01 in the presence of serial dilutions of the recombinant proteins and incubated for 3 days at 37 °C. Then, cell viability was determined using crystal violet staining as a measure of the extension of cytopathic effect; cells were fixed with 10% formaldehyde, stained with crystal violet solution (20% Ethanol, 0.1% Crystal Violet), and after washing, the absorbance at 595 nm was recorded for each well in a spectrophotometer. Assays were conducted at least in triplicates and the inhibitory concentrations 50 (IC50) for each protein were estimated by a nonlinear regression fitting of the data as the protein concentration necessary to reduce cytopathic effect on MDBK cells by 50% compared to control infected and non-treated cells.

### 2.8. Selection of Revertant Viruses

MDBK cells were seeded in 24 well plates, transfected with in vitro transcribed RNA of mutant BVDV and incubated under 5% CO⁠_2_ at 37 °C for 3 days. Then supernatants were collected, and cells lifted with trypsin and re-seeded in 24 well plates. After each cell passage, RNA was extracted from the supernatant and a fragment corresponding to E2 was amplified by RT-PCR using specific primers 1 (5′-GACCCGGGACACTTGGATTGCAAACC-3′) and 7 (5′-TAACTGCAGGTGATGGTCAGTCAC-3′).

## 3. Results

### 3.1. Design of Mutations

To investigate the structure-function relationship for the exposed β-hairpin of E2 domain II, we designed two amino acid changes. Amino acid conservation analysis of the 12-residue stretch comprising this motif within pestiviruses shows an overall conservation of 65%, with a conserved polar residue (Ser or Asn) at position 144 and strictly conserved Thr and Leu residues at positions 147 and 148 (Figure 1). Notably, neutralization escape variants did not mutate at these conserved residues, and substitution analysis showed that Ser to Asn mutation at position 144 in the context of CSFV does not impact virus infectivity [18,21]. Thus, in order to study whether this β-hairpin motif is involved in the interaction of E2 from BDVD with cell receptors, we decided to mutate polar residues Asn 144 and Thr 147 to Ala in the context of BVDV E2.

### 3.2. Mutation of the β-Hairpin Motif Impairs Function of E2

To study the impact of mutations in the β-hairpin motif, we designed biotin labeled wild type and mutant BVDV E2 recombinant proteins using baculoviruses. To this end, we developed a system to achieve site specific biotinylation in insect cells. First, we constructed recombinant baculoviruses carrying an expression cassette for wild type or mutant BVDV E2, consisting of baculovirus gp64 signal sequence fused to the biotin acceptor peptide and a truncated version of E2 tagged with 6xHis (Figure 2A). In turn, Sf9 cells were transfected with an insect expression plasmid encoding *E. coli* biotin ligase BirA fused to a signal sequence that directs the protein to the secretory compartment [25], and BirA expressing cells were established following antibiotic selection. To confirm specific biotinylation of E2, parental cells or BirA expressing cells were infected with recombinant baculovirus in cell culture media supplemented with biotin, and protein expression in the cell culture supernatant was assessed by Western blotting with HRP-conjugated streptavidin and anti-6xHis antibody (Figure 2B). As expected, a band of apparent molecular weight of 53 kDa was detected in the supernatant of both cell lines with anti 6xHis antibody, but biotinylation as detected by streptavidin-HRP reactivity occurred only in BirA expressing cells (Figure 2B). Biotinylated wild type and mutant E2 recombinant proteins were purified by Ni-NTA affinity chromatography and recombinant protein identity confirmed with a specific anti E2 polyclonal antibody that showed comparable reactivity against both proteins (Figure 2C).

Different lines of evidence indicate that pestivirus E2 mediates receptor recognition. Therefore, we reasoned that mutations in the β-hairpin might interfere with binding of E2 to the surface of BVDV susceptible cells. We evaluated binding of the soluble proteins to susceptible cells by Western blotting and flow cytometry. To this end, MDBK cells grown in monolayer were incubated with 1 or 0.2 µg of recombinant proteins, and after extensive washing to remove non-bound protein, the amount of protein that remained bound to the cell monolayer was measured by two different methods. On the one hand, total cell extracts were harvested in gel loading buffer for SDS-PAGE, and the presence of E2 was detected by Western blot using an anti-E2 polyclonal antibody. On the other hand, cells were lifted from the plate with EDTA and immunostained with an anti E2 specific polyclonal antibody for flow cytometry analysis. In agreement with the notion that the β-hairpin mediates engagement with receptors, we observed that binding of the mutant protein to MDBK cells was diminished compared to wild type. The relative intensity of the band corresponding to bound E2 protein was reduced in Western blots (Figure 3A), and the flow cytometry histogram was shifted to lower intensities for mutant E2 (Figure 3B). As an approach to test the specificity of binding, we compared binding of wild type protein to MDBK cells and to cell lines non-susceptible to BVDV infection. We used CRIB cells, which are derived from MDBK cells and are resistant to infection due to a specific block of BVDV entry [29,30], and HeLa cells of human origin [13]. Flow cytometry measurements of binding showed that wild type E2 only bound MDBK cells (Figure 3D), suggesting that binding is mediated by recognition of specific receptors on the surface of susceptible cells.

To further investigate the impact of mutations in the β-hairpin on E2 function, we measured the ability of recombinant proteins to inhibit BVDV infection in a cytopathic effect reduction assay [26]. Consistent with previous results [10], treatment of MDBK cell monolayers with wild type E2 prior to infection resulted in a dose-dependent inhibition of BVDV infection that reached 100% at 1 µg/mL. The effective concentration 50 (EC_50_) for wild type E2 was 0.02 µg/mL. Treatment with the mutant protein also yielded 100% inhibition at 1 µg/mL, but the inhibition curve was shifted towards higher protein concentrations resulting in an EC_50_ of 0.1 µg/mL, thus reflecting decreased function of the mutant protein to inhibit BVDV infection. As the ability of E2 to inhibit virus infection relates to receptor binding on the cell surface, together with Western blot and flow cytometry analyses, these results indicate that mutations in the β-hairpin impair E2 function.

### 3.3. The E2 β-Hairpin Is Critical for BVDV Infection

We next sought to assess the role of the β-hairpin in E2 in the context of a BVDV infectious particle. To this end, we introduced N144A/T147A mutations into the cDNA of non-cytopathic BVDV infectious clone. Equal amounts of wild type and mutant viral RNAs transcribed in vitro were transfected into MDBK cells (Figure 4A) and expression of E2 was evaluated by immunofluorescence staining of transfected cells (Figure 4B). At 24 h after transfection, the levels of expression of E2 were similar in cells transfected with wild type and mutant RNAs, suggesting that mutation of the β-hairpin did not impact on virus RNA replication. Virus production to the cell culture supernatant was measured by qPCR (Figure 4C). RNA extracted from the clarified supernatant of transfected cells was the template for cDNA synthesis, and the number of genome copies quantified by qPCR was used as an estimate of BVDV titers. Transfection with wild type and mutant RNAs yielded comparable virus titers, indicating efficient assembly into BVDV particles of the mutant RNA. Finally, to test infectivity of the recovered virus stocks, we used the supernatants of transfected cells to infect a fresh monolayer of MDBK cells, and BVDV infection was evaluated by immunostaining against E2 at 48 h. As expected, the wild type virus was able to infect and propagate in MDBK cells as evidenced by the formation of foci of infection. In contrast, the infectivity of the mutant virus was compromised, as the number of immunostained cells was only marginal. Together, these results indicate that N144A/T147A mutations in the exposed β-hairpin of E2 domain II affect virus infectivity underscoring a role for this motif in BVDV envelope protein function.

### 3.4. BVDV Infectivity Is Recovered after Reversion of N144A and T147A Mutations

To further confirm the association between N144A and T147A mutations and loss of E2 function, we pursued experimental passaging of the mutant virus in cell culture. In vitro transcribed RNA of the mutant virus was transfected into MDBK cells, and every 3 days the supernatant was collected to assess the infectivity of the released virus particles and cells were lifted and passaged. Infectivity was evaluated by immunofluorescence staining against E2 after reinfection of a fresh monolayer of cells with the supernatants recovered after every cell passage (Figure 5). The supernatant recovered at 3 days after transfection (Passage 0) yielded a small number of individual cells that were positive for E2. At passage 1, discrete foci of infection were detected, and after four passages, virus supernatants were able to infect and spread in a fresh monolayer of cells. Sanger sequencing of E2 from viruses recovered in cell supernatants showed that viruses in the passage 0 stock retained the N144A and T147A mutations. Next, sequence analysis revealed a mixed population of viruses carrying the mutated (E2-A147) and wild type (E2-T147) residue at position 147 in the stock of passage 1. In passage 4, virus population had reverted to wild type residues (E2-N144 and T147). With respect to the sequence of passage 0 virus population, a single additional mutation was found in the E2 sequence of the revertant virus population that resulted in the change of Pro to Ser at residue position 285 in domain III, which is the natural amino acid in the reference NADL sequence. Altogether, these results suggest that regain of infectivity can be attributed to reversion of the mutated residues to wild type.

## 4. Discussion

The role of Pestivirus E2 as a determinant of virus tropism is widely accepted and is linked to the ability of E2 to engage cellular receptors. However, direct functional mapping of the motifs involved in virus-receptor interaction and a comprehensive characterization of entry receptors have remained elusive. To guide the mapping of receptor binding motifs, we took a reverse genetics approach. Based on sequence conservation among Pestiviruses, protein structure and published functional analyses of a previously recognized linear epitope in E2, we defined a β-hairpin motif and designed mutations at two residues positions in BVDV E2 to disrupt receptor binding by changing polar to non-polar amino acids. 

For CSFV, extensive mutational analysis of this motif has been performed, revealing that the β-hairpin plays a critical role in virus replication. Substitution of residues encompassing the loop in the β-hairpin motif of CSFV E2 with the homologous residues from BVDV resulted in a modest reduction in virus replication in porcine cells [18,19]. In contrast, substitutions extending towards the adjacent β-sheet had a progressive impact on virus replication in porcine cells and resulted in up to undetectable virus yields. It is possible that the latter set of substitutions disrupted protein conformation affecting receptor binding domains in E2 and abolished growth in cell culture. Here, we used recombinant soluble proteins to directly evaluate the impact of mutations on E2 function. Cell binding assays showed that selected residue mutations N144A and T147A reduced the ability of BVDV E2 to bind bovine susceptible cells. Furthermore, we observed that binding was specific to BVDV susceptible cells. Protein functionality was further evaluated using wild type and mutant E2 to compete BVDV infection. The fact that the mutant protein had a higher EC_50_ but was still able to completely inhibit infection at high doses supports the notion that the designed mutations do not have a major impact on protein conformation. However, more extended biochemical characterization of wild type and mutant proteins is required to rule out the possibility that selected mutations interfered with protein folding causing an indirect effect on receptor binding.

In agreement with the presumed role of this β-hairpin motif in virus entry/receptor recognition, we found that upon transfection of a genome carrying N144A and T147A mutations, viral particles are produced to the cell culture supernatant to wild type levels, but these viruses have impaired ability to infect susceptible cells. Different from previous approaches that addressed the analysis of this motif by constructing chimeras of CSFV and BVDV [18,19], we mutated conserved polar amino acids across the Pestivirus genus to Ala. These substitutions would result in a change in the electrostatic potential of the loop in the β-hairpin motif and thus disrupt protein–protein interactions; this is BVDV receptor binding. Furthermore, the mutant virus only regained infectivity upon reversion of the mutated amino acids to wild type, and no compensatory mutations were found in E2, narrowing the function of receptor binding to this β-hairpin motif. So far, CD46 is the only identified receptor for BVDV. It is well established that treatment of MDBK cells with monoclonal antibodies against CD46 can block infection [10,11]. Interestingly, polarized bovine respiratory epithelial cells only express CD46 on the apical side, and accordingly, treatment with monoclonal antibodies against CD46 blocked BVDV infection from the apical surface [31]. However, respiratory epithelial cells were efficiently infected by BVDV from the basolateral side despite the absence of CD46 at the basolateral surface, suggesting a CD46 independent entry mechanism. Further supporting this alternative mechanism of entry, generation of stable CD46-knockout MDBK cells showed that BVDV was still able to enter into knockout cells, albeit with markedly reduced growth [14]. In turn, the role of CD46 as a receptor for different pestiviruses of swine is controversial [13,15]. Porcine cell lines that do not express CD46 were shown to be non-permissive to APPV infection, while porcine pestiviruses such as CSFV and BuPV efficiently infected these cells, indicating a CD46-independent mechanism of entry [15]. Additional binding partners of CSFV E2 have been identified and mapping of amino acids involved in these interactions has been achieved using yeast-two hybrid system [32,33,34]. These factors do not appear to play a role in virus entry, and the regions involved in these interactions fall outside the β-hairpin motif in domain II. The present study was limited to the characterization of the role of this β-hairpin motif in virus-receptor interaction using a directed mutagenesis approach. Our results rise interesting questions regarding the contribution of individual residues to receptor binding and the role of this motif as a determinant of cell tropism. For instance, N144 is linked to D146 in Pestiviruses A, B and G that share in vitro tropism to MDBK cells, while S144 is linked to either S146 or T146 in the remainder of the Pestiviruses. Hence it is possible to speculate that there could be an interdependency between these two residues. On the other hand, the interaction of wild type and mutant soluble E2 with CD46 was not directly investigated here. Besides, our attempts to identify BVDV E2 ligands using the biotinylated proteins as a bait in pull-down assays were not successful. Altogether, identification of the binding partner of this β-hairpin motif warrants further investigation.

Finally, we were able to adapt baculovirus expression vector system to site specific biotinylation of BAP-tagged proteins. The system was based on the development of a novel Sf9 cell line expressing biotin ligase BirA and allows efficient biotin coupling with the only requirement of the construction of recombinant baculoviruses carrying an expression cassette for the heterologous protein fused to the 15-amino acid BAP. The system can serve as a tool for production of recombinant proteins with different applications including the design of diagnostic test that can use conjugated streptavidin for detection.

In conclusion, our structure–function analysis provides new insights into the molecular determinants of virus-receptor interactions for BVDV, narrowing receptor binding function to a 12-residue stretch of conserved amino acids in a β-hairpin motif in domain II of Pestivirus E2.

## Figures and Tables

**Figure 1 viruses-13-01157-f001:**
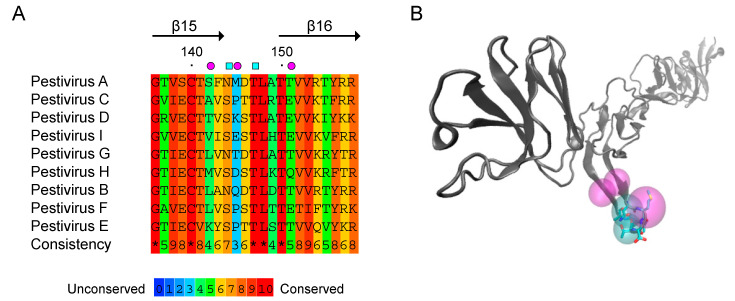
Conservation analysis of amino acids encompassed in the solvent exposed β-hairpin motif of BVDV E2. (**A**) Amino acid alignment of the β-hairpin motif of Pestivirus E2. PRALINE [27] multiple sequence alignment method was used to calculate amino acid conservation. Genebank Accession numbers for Pestivirus sequences were retrieved from [1]. Pestivirus A (BVDV-1/M96751), Pestivirus B (BVDV-2/AF002227), Pestivirus C (CSFV/AF326963), Pestivirus D (BDV/AF037405), Pestivirus E (pronghorn pestivirus/AY781152), Pestivirus F (BuPV/EF100713), Pestivirus G (giraffe pestivirus/AF144617), Pestivirus H (Hobi-like pestivirus/FJ040215), Pestivirus I (Aydin-like pestivirus/JX428945). Magenta circles indicate residues involved in antibody escape in CSFV [21] and cyan squares the amino acids mutated to Ala in this study. (**B**) Cartoon representation of BVDV E2 fold (PDB 4JNT) was prepared with VMD software [28]. Mutated amino acids Asn144 and Thr147 are depicted as stick representations. Cyan and purple spheres correspond to the shapes shown in (**A**).

**Figure 2 viruses-13-01157-f002:**
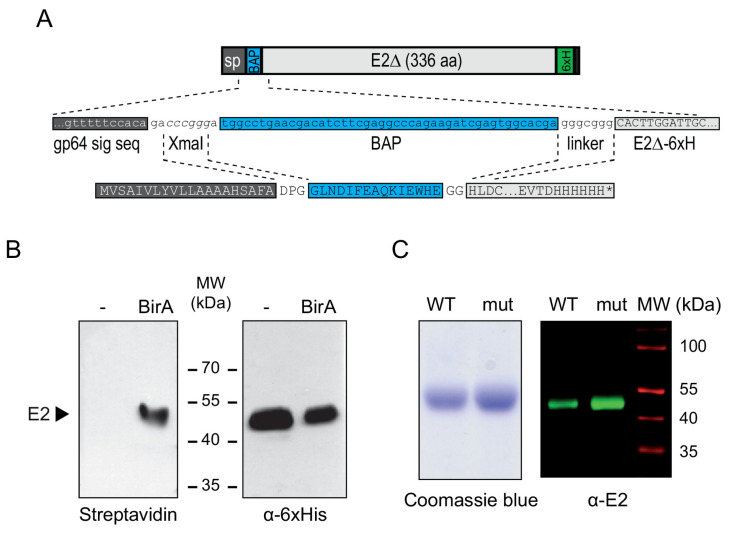
Design of a biotinylated version of BVDV E2. (**A**) Schematic representation of the secreted version of BVDV E2 expressed in insect cells. C-terminally truncated E2 (E2∆) was fused to baculovirus GP64 signal peptide (sp) and biotin acceptor peptide (BAP) sequences at the N-terminus and to a 6xHis tag (6xH) at the C-terminus. (**B**) Site specific biotinylation of E2 in cells expressing *E. coli* biotin ligase BirA. The supernatants of parental or BirA expressing Sf9 cells infected with the recombinant baculovirus bearing BAP-E2 expression cassette were resolved in SDS-PAGE and subjected to Western blotting with HRP-conjugated streptavidin (left) or a polyclonal antibody against 6xHis (right). (**C**) Expression of wild type and mutant E2 in insect cells. Coomassie Blue staining (left) and Western blot against E2 (right) of the purified proteins resolved by SDS-PAGE.

**Figure 3 viruses-13-01157-f003:**
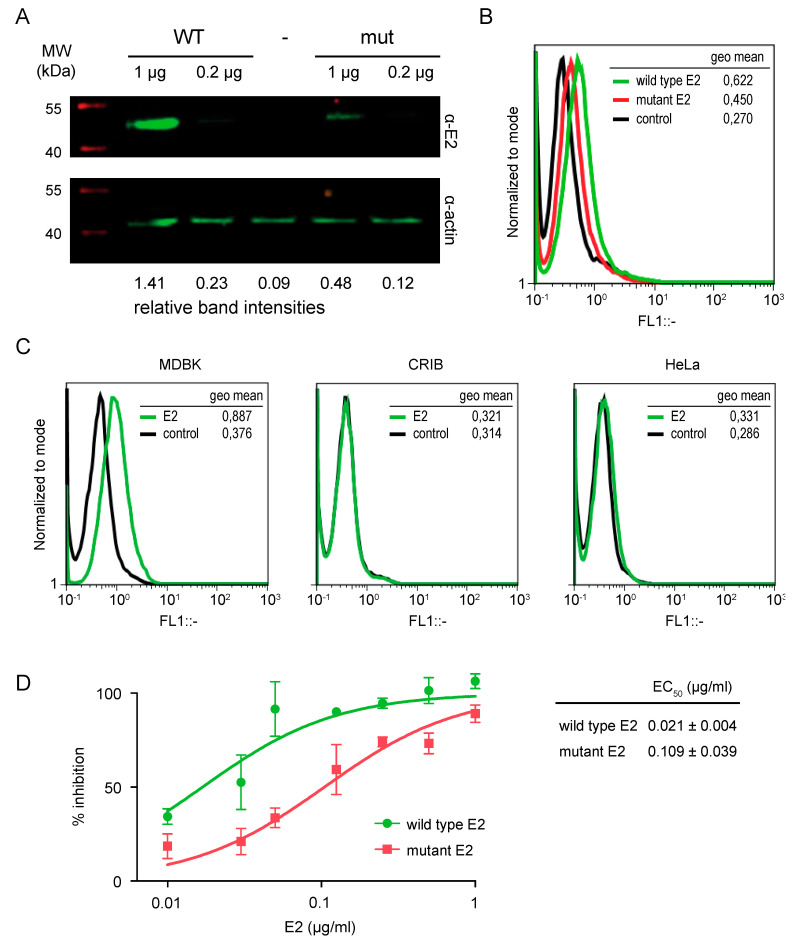
Mutation of the exposed β-hairpin motif impairs E2 function. (**A**,**B**) Binding of wild type and mutant E2 soluble form to the surface of BVDV susceptible MDBK cells detected by Western blot and flow cytometry. (**A**) Western Blot. Bound E2 was detected in cell lysates resolved by SDS-PAGE using an anti E2 polyclonal antibody. Actin was used as a loading control. Relative band intensities are indicated for each lane. The gel is representative of three independent experiments. (**B**) Flow cytometry. Bound E2 was detected by surface staining of fixed samples with a polyclonal antibody against E2. Histogram of relative fluorescence intensity (FL1) for control cells (control, black line), and cells treated with wild type (green line) or mutant (red line) E2. The histogram is representative of two independent experiments. (**C**) Specificity of binding. Binding of wild type soluble form of E2 to the surface of MDBK cells and non-susceptible CRIB and HeLa cells detected by flow cytometry. Histogram of relative fluorescence intensity (FL1) for control cells (control, black line), and cells treated with wild type E2 (green line). Fluorescence intensities recorded as the geometric mean of the cell population are presented. (**D**) Functionality of recombinant E2 tested in a cytopathic effect reduction assay. MDBK cells were preincubated with increasing amounts of the recombinant protein and then infected with cpBVDV at an MOI of 0.01. At 3 to 4 days postinfection, cells were fixed and stained with crystal violet to estimate the extent of the cytopathic effect. Plot of the log concentration versus the percentage of inhibition. Error bars represent the standard deviation of triplicates for each point. Inhibitory concentrations were estimated from nonlinear regression fitting of the curve with GraphPad Prism 5 software. EC_50_ values are the mean and standard deviation from three independent experiments.

**Figure 4 viruses-13-01157-f004:**
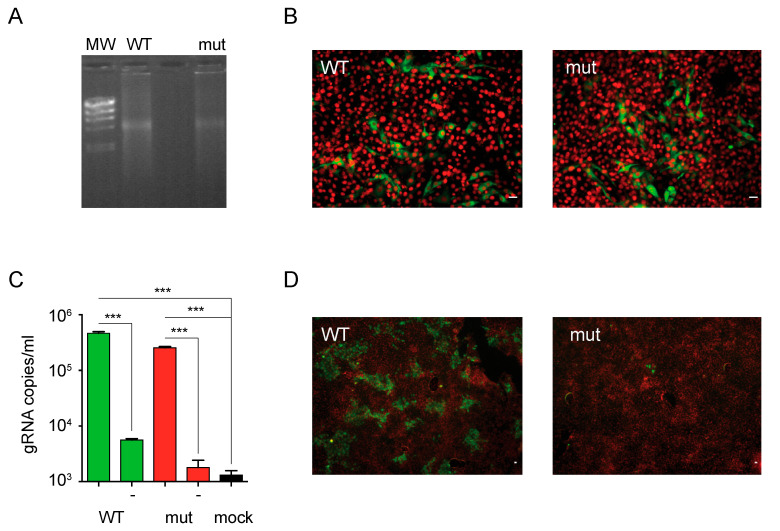
BVDV particles bearing mutations in the exposed β-hairpin motif display reduced infectivity. (**A**) Agarose gel electrophoresis of in vitro transcribed genomic RNAs of wild type (WT) and mutant (mut) ncpBVDV. Lambda DNA/HindIII marker (MW) was used as a reference. (**B**) E2 expression in cells transfected with wild type and mutant RNAs. Representative fluorescence microscopy images acquired with 20× objective of MDBK cells transfected with wild type (WT) or mutant (mut) RNAs and stained with a polyclonal antibody against E2 (green channel) and DAPI (red channel). Scale bar: 10 µm. (**C**) BVDV virion production to the supernatant of transfected cells. Bar graph for the titers of wild type and mutant viruses produced to the supernatant of transfected cells estimated by qPCR. No reverse transcriptase controls are indicated with (-). The black bar represents a control for mock transfected cells. Error bars represent the standard deviation of triplicate points. Data were analyzed by one-way ANOVA with Bonferroni’s posttest (***, *p* < 0.001). (**D**) Infectivity of wild type and mutant BVDV virions. Representative fluorescence microscopy images acquired with 4× objective of MDBK cells infected with wild type (WT) or mutant (mut) BVDVs recovered from transfected cells and stained with a polyclonal antibody against E2 (green channel) and DAPI (red channel). Scale bar: 10 µm. The results are representative of at least three independent transfections.

**Figure 5 viruses-13-01157-f005:**
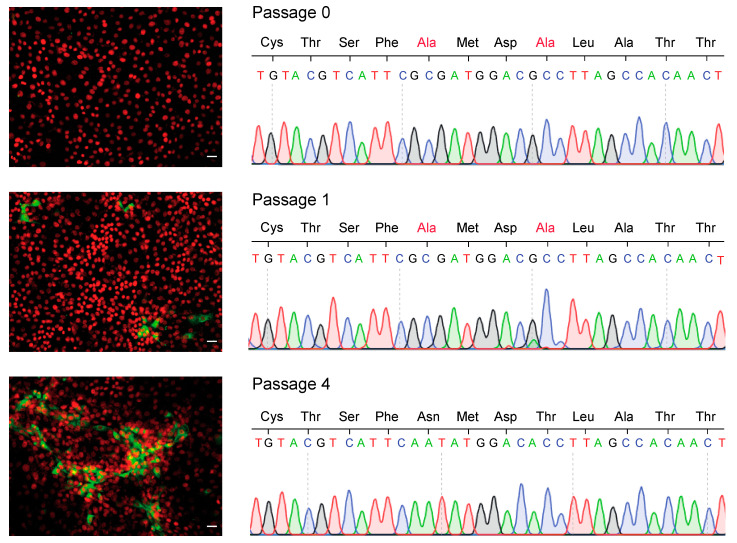
Reversion of mutations in the exposed β-hairpin motif restores BVDV infectivity. MDBK cells were transfected with the genomic RNA of the mutant virus and passaged every 3 days. The supernatant of transfected cells (Passage 0) and after one (Passage 1) or four passages (Passage 4) was used to infect fresh MDBK cells to follow the recovery of infectious viruses. (Left) Representative fluorescence microscopy images acquired with 10× objective of infected MDBK cells stained with a polyclonal antibody against E2 (green channel) and DAPI (red channel). Scale bar: 10 µm (Right) Sequence chromatogram of the 12 amino acid β-hairpin motif of viruses recovered in the cell culture supernatant at passages 0, 1 and 4.

## Data Availability

Data is contained within the article.

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
