# Peer review of "A β-Hairpin Motif in the Envelope Protein E2 Mediates Receptor Binding of Bovine Viral Diarrhea Virus"

_viruses, 2021, doi:10.3390/v13061157_

Round 1
Reviewer 1 Report
Despite the well-known implication of the pestiviruses E2 glycoprotein in virus-host interaction mechanisms, in the generation or evasion of the host's immune response and the important role virus replication, the E2 residues involved in the interaction of the virus and its receptors that enter the cells remain as gaps and should still be extensively studied.
In this work, the authors contribute to broadening the knowledge on these fundamental aspects of Pestivirus-host interaction. The authors demonstrate with a solid methodology that the β-hairpin conserved region of the E2 protein is critical for the interaction with host cell receptors. The methodology established in this work will allow increasing knowledge in these aspects. The results have been well written and supported with good figures.
The baculovirus system methodology used in this work opens the possibility of generating recombinant proteins of great value in their application for virus-host interaction studies, and even that will be of great value in the design of diagnostic strategies.
I have no doubt as to the results obtained.
Regarding the discussion of the work, the authors correctly use the current bibliography, however, the discussion, although it can be continued, is a bit simple, being in some points little careful in terms of writing. I recommend that the authors review the general wording of the discussion, especially lines 344-362.
Author Response
Regarding the discussion of the work, the authors correctly use the current bibliography, however, the discussion, although it can be continued, is a bit simple, being in some points little careful in terms of writing. I recommend that the authors review the general wording of the discussion, especially lines 344-362.
Following the reviewer suggestion, we the Discussion has been extended to deepen into the link of E2 and Pestivirus receptor usage and discuss the limitations of the study in this context. Also, lines 344-362 have been reworded.
Reviewer 2 Report
This is a somewhat limited mutational investigation on just two amino acids of the BDVD E2 protein. The results show that the combined mutation of N144 and T148 in the E2 result in a primarily non viable BVDV recombinant virus that reverts to its original sequence within 4 passages. In an inhibition experiment with insect cell expressed E2 the Asn144/T148 mutant was 5 fold less efficient compared to the wt E2.
Strenghts:
- very well written manuscript and thorough experiments. Most controls are in place and the results are clearly demonstrated. Interpretations are sound.
- timely and relevant topic.
- The biochemical characterisation of the soluble E2 molecules and the inhibition experiments are elegant.
- Referring to data from the CSFV field is very useful and helpful, because often the "other" virus system in disregarded. The analogous site of your mutations is the well characterised SPTTLR motif which makes a potent epitope for neutralising antibodies.
- Along these lines there is an almost prehistoric publication (1997) (Hulst MM, Moormann RJ. Inhibition of pestivirus infection in cell culture by envelope proteins E(rns) and E2 of classical swine fever virus: E(rns) and E2 interact with different receptors. J Gen Virol. 1997 Nov;78 ( Pt 11):2779-87. doi: 10.1099/0022-1317-78-11-2779. PMID: 9367363.) that on the one side introduced the inhibition of infection with soluble glycoproteins and on the other show that a CSFV E2 blocks the infection with BVDV.
Weaknesses
- why were only these two amino acids picked for mutation? The loop is much larger and a random mutagenesis approach would from my point of view be more informative. Have you tried 1) single mutants and 2) deletions instead ob substitutions?
- The binding experiment of BIR A biotinylated E2 to MDBK cells is informative but it would be necessary 1) to include other cells that are not infected by BVDV such as human or mouse cells and 2) to combine it with neutralising antibody incubations (e.g anti E2 , anti CD46). Why was BIR A biotinylation used anyway, neither purification nor the analyses made use of it.
Recommendations
- Why didn't you express and purify the fluorescently labelled E2 you presented in a previous paper instead of biotinylated-E2? These assays would be so much easier to do.
- The amino acid sequence alignment in the named region (Fig.1) reveals that N144 always goes along with D146 (in Pestivirus A,B and G). Hence one could speculate that there is a interdependecy between these two residues. It would be interesting to change not only N144A but also D146S or N144S and D146S.
Author Response
why were only these two amino acids picked for mutation? The loop is much larger and a random mutagenesis approach would from my point of view be more informative. Have you tried 1) single mutants and 2) deletions instead of substitutions?
We acknowledge the limitations of the current study and they are discussed in the revised manuscript (lines 454-464). The rationale for the design of mutations is presented in section 3.1, and our approach was based on targeting of conserved residues. Unfortunately, we did not test the effect of single mutations or deletions.
The binding experiment of BIR A biotinylated E2 to MDBK cells is informative but it would be necessary 1) to include other cells that are not infected by BVDV such as human or mouse cells and 2) to combine it with neutralising antibody incubations (e.g anti E2, anti CD46). Why was BIR A biotinylation used anyway, neither purification nor the analyses made use of it.
As an approach to show the specificity of binding of E2 to MDBK cells, we included experiments showing that the wild type protein does not bind non-susceptible CRIB cells of bovine origin or HeLa cells (Figure 3D). We recognize the value of experiments with neutralizing antibody incubations. However, in our hands these experiments were technically complicated because we achieved the most sensitive detection of binding with a mouse polyclonal antibody against E2, and the antibodies against E2 and CD46 were also raised in mouse. Finally, we originally designed biotinylated E2 to be used as a bait in pull-down experiments to identify novel E2 partners (see lines 462-463).
Why didn't you express and purify the fluorescently labelled E2 you presented in a previous paper instead of biotinylated-E2? These assays would be so much easier to do.
As suggested by the reviewer, a fluorescently labeled version of E2 would have been useful to detect binding in flow cytometry assays. In our previous work, we reported fluorescently labeled E2 that was expressed in the context of the BVDV genome, not as a soluble protein.
The amino acid sequence alignment in the named region (Fig.1) reveals that N144 always goes along with D146 (in Pestivirus A, B and G). Hence one could speculate that there is a interdependecy between these two residues. It would be interesting to change not only N144A but also D146S or N144S and D146S.
This is an interesting observation that we did not explore. The possible interdependence between N144 and D146 is discussed in the revised manuscript.
Reviewer 3 Report
In the study under review, Merwaiss and colleagues are investigating the mapping of the E2 ectodomain to further characterize the putative interactions with the cellular receptor of BVDV. Studies on this ß-hairpin in CSFV have been presented earlier. They were able to establish a structure-function relationship for a β-hairpin loop in the E2 by altering two single amino acids reversing the electrostatic potential of the loop. A recombinantly produced soluble E2 harboring these mutations showed reduced binding to host cells and a weaker block of BVDV infection compared with wild type E2. In particles of recombinantly produced BVDV, the mutations also inhibited the ability of the virus to infect susceptible cells. I really liked this study because it takes a coherent approach, presents controlled experiments, and is very well presented.
Major points:
- CD46 has been initially characterized as a receptor molecule for BVDV and has been confirmed by many studies. Therefore, as a criticism of Merwaiss' study, however, it must be mentioned that a direct investigation of the ligand-receptor molecule interaction is missing. Please exlain, if it was either not investigated or did not lead to success. From this point of view, the reviewer would have liked to see a broader discussion that more strongly emphasizes the limitations of the study. The possibility that the mutations "only" disrupt the structure of the E2 molecule and lead to misfolding causing an indirect effect on receptor binding should be discussed.
- Overall, the reviewer would have liked to see a broader discussion. This discussion mainly recapitulates the results presented and does not really put it in the context of recent results on pestiviral receptor and co-receptor molecules. In the last few months alone, several new papers have appeared on CSFV and APPV receptors that should be discussed in the context. Therefore, I would like to ask the authors to still include this in the discussion.
Minor points:
- Line 37ff: The initial letters of the old species names are written in lower case, so it must be bovine viral diarrhea virus, classical swine fever virus, and border disease virus. For the other species, it is spelled correctly.
- 2, B and C: The reviewer does not understand why only the small sections of the gels and Western blots are shown, since a complete gel image in C should be used to assess the purity of the protein preparations and in B and to assess the specificity of the labeling.
- 3A: An RNA size standard is missing! If no RNA standard is available, at least a DNA ladder should be used as a workaround, so that comparability is given and the reader can estimate the molecular size. As mentioned above, a complete presentation of the gel image would also be helpful to better assess the synthesis quality.
- 3B and D and Fig. 5: A size bar is missing to give the reader a comparison. This is especially important with the different magnification in the two pictures in a single figure seen in Fig. 3, as otherwise the reader would have to trudge through the entire figure legend to find the magnification. I also hope that the quality of the figures in the final manuscript is higher than in the PDF for the review, because I could hardly recognize anything.
Overall, however, I would like to congratulate the authors on their manuscript and can only welcome a publication of the clear results after incorporation of the minor change requests.
Author Response
Major points:
CD46 has been initially characterized as a receptor molecule for BVDV and has been confirmed by many studies. Therefore, as a criticism of Merwaiss' study, however, it must be mentioned that a direct investigation of the ligand-receptor molecule interaction is missing. Please explain, if it was either not investigated or did not lead to success. From this point of view, the reviewer would have liked to see a broader discussion that more strongly emphasizes the limitations of the study. The possibility that the mutations "only" disrupt the structure of the E2 molecule and lead to misfolding causing an indirect effect on receptor binding should be discussed.
We acknowledge the limitations of the study pointed out by the reviewer that are now mentioned in the Discussion of the revised manuscript (lines 424-426 and 461-464). Our attempts to characterize CD46 as the ligand for the hairpin motif led to non conclusive results. On the one hand, wild type protein did not bind CRIB cells of bovine origin that were reported to express functional CD46. On the other hand, in line with previous reports indicating that labeled BVDV virions bind to heterologous cells expressing bovine CD46, upon overexpression of CD46 in BHK cells of hamster origin we observed a modest increase in wild type protein binding over background binding of the mutant protein or binding of the wild type protein to parental BHK cells.
Overall, the reviewer would have liked to see a broader discussion. This discussion mainly recapitulates the results presented and does not really put it in the context of recent results on pestiviral receptor and co-receptor molecules. In the last few months alone, several new papers have appeared on CSFV and APPV receptors that should be discussed in the context. Therefore, I would like to ask the authors to still include this in the discussion.
Following the reviewer suggestion, we included a passage discussing Pestivirus dependency on CD46 to enter host cells in the Discussion and cited the recent work published for CSFV, APPV and BVDV (lines 438-450).
Minor points:
Line 37ff: The initial letters of the old species names are written in lower case, so it must be bovine viral diarrhea virus, classical swine fever virus, and border disease virus. For the other species, it is spelled correctly.
Writing was corrected in the text.
2, B and C: The reviewer does not understand why only the small sections of the gels and Western blots are shown, since a complete gel image in C should be used to assess the purity of the protein preparations and in B and to assess the specificity of the labeling.
Gel sections were replaced by complete images.
3A: An RNA size standard is missing! If no RNA standard is available, at least a DNA ladder should be used as a workaround, so that comparability is given and the reader can estimate the molecular size. As mentioned above, a complete presentation of the gel image would also be helpful to better assess the synthesis quality.
The original image was replaced by the complete presentation of a gel showing the migration of wild type and mutant RNAs next to a DNA ladder.
3B and D and Fig. 5: A size bar is missing to give the reader a comparison. This is especially important with the different magnification in the two pictures in a single figure seen in Fig. 3, as otherwise the reader would have to trudge through the entire figure legend to find the magnification. I also hope that the quality of the figures in the final manuscript is higher than in the PDF for the review, because I could hardly recognize anything.
We apologize for the quality of the images that was probably an issue resulting from the assembly of the manuscript for peer review. Scale bars were included in the micrographs of Figures 3 and 5 as required by the reviewer.
Reviewer 4 Report
In the presented manuscript, Merwaiss et al. communicate their findings on two specific mutations in an exposed beta-hairpin of the BVDV E2 protein on the ability of the virus to infect new host cells. They test the relevance of these two amino acids in the context of a blocking experiment with recombinantly expressed E2 and by introducing the mutations in a full length clone of the NADL strain. The findings are in the opinion of this reviewer relevant and interesting.
Please find below questions that came to mind whilst reading your manuscript and also some suggestions.
A) In the text and the material and methods section, both, polyclonal antibody and antibody, are used. Does this always refer to the polyclonal antibody? Please clarify.
B) Why are titers evaluated by qPCR only and not by assessing the amount of infectivity in the supernatant? Also, can you be sure that your DNaseI step is sufficient to result in complete elimination of template DNA? In the same line of thought, why are the E2 inhibitory doses not calculated based on qPCR results but rather by evaluation of the extent of the cpe? What is your calculation method to quantify the cpe? Same for the quantification of growth of the revertant. I am missing a quantification of virus output in comparison to wt here, especially if you state that 'infectivity was restored'.
C) Fig. 1B, could you mention in the legend that the cyan and purple spheres are corresponding to the shapes shown in 1A? Well, at least if the way I understood this is correct.
D) Could you comment why the binding assay was performed at room temperature? Also, in order to get a clearer picture of the results of your binding assays, could you please quantify the results of the flow cytometry and the Western blot.
E) Could you please state for all of your experiments how often they were replicated and whether it was biological replicates or done in the same experimental run.
F) Fig. 2, what are the funny symbols in A referring to? In the legend, could you please either write C-terminus and N-terminus or introduce the abbreviation?
G) Regarding the biotin ligase, I find this a very intriguing system. Could you please comment on where exactly the enzyme is located in the insect cells or whether it is also being secreted. And, as I did not understand this, why do you need the biotinylation in the context of E2 purification? As far as I understand, you are purifying with the His-tag?
H) Fig 4 C, the black bar in the negative control, does it refer to your detection limit or is it a contamination, or something else? Could you please clarify this?
I) In Fig 5 under passage 4, do you think it is possible to adjust the grey vertical dashed lines so that they are at the same nucleotide as for the other two passages? I think this would make it easier for the reader.
Author Response
A) In the text and the material and methods section, both, polyclonal antibody and antibody, are used. Does this always refer to the polyclonal antibody? Please clarify.
A mouse polyclonal antibody against E2 previously generated by us was used across the study. To avoid confusion, this antibody is now referred to as polyclonal antibody.
B) Why are titers evaluated by qPCR only and not by assessing the amount of infectivity in the supernatant? Also, can you be sure that your DNaseI step is sufficient to result in complete elimination of template DNA? In the same line of thought, why are the E2 inhibitory doses not calculated based on qPCR results but rather by evaluation of the extent of the cpe? What is your calculation method to quantify the cpe? Same for the quantification of growth of the revertant. I am missing a quantification of virus output in comparison to wt here, especially if you state that 'infectivity was restored'.
We chose to estimate virus titers by qPCR because mutations in the beta-hairpin compromised infectivity and hence precluded the possibility of tittering using infectivity-based assays. Bars corresponding to quantification for reaction mixes without RT were included in the graph and show that the DNaseI step was efficient in removing DNA. The cytopathic effect reduction assay is routinely used in our laboratory. A paragraph describing the corresponding method was included in the Materials and Methods section to clarify the procedure. Finally, as pointed out by the reviewer infectivity of the revertant virus population was not compared with wild type virus. To address this point we toned down our statement (lines 365-366): “At passage 1, discrete foci of infection were detected, and after four passages virus supernatants were able to infect and spread in a fresh monolayer of cells”.
C) Fig. 1B, could you mention in the legend that the cyan and purple spheres are corresponding to the shapes shown in 1A? Well, at least if the way I understood this is correct.
The Figure legend was modified accordingly.
D) Could you comment why the binding assay was performed at room temperature? Also, in order to get a clearer picture of the results of your binding assays, could you please quantify the results of the flow cytometry and the Western blot.
During optimization of the binding assays we observed that detection of surface-bound protein in non-permeabilized cells by flow cytometry was more sensitive using room temperature incubation rather than incubation on ice. Geometric mean of fluorescence intensity for cell populations in flow cytometry experiments and relative intensity of bands in Wetern blot assays were included in Figure 3.
E) Could you please state for all of your experiments how often they were replicated and whether it was biological replicates or done in the same experimental run.
The number of replicates in an experimental run and biologically independent replicates of all the experiments are now stated in the manuscript.
F) Fig. 2, what are the funny symbols in A referring to? In the legend, could you please either write C-terminus and N-terminus or introduce the abbreviation?
A new Figure 2 was assembled
G) Regarding the biotin ligase, I find this a very intriguing system. Could you please comment on where exactly the enzyme is located in the insect cells or whether it is also being secreted. And, as I did not understand this, why do you need the biotinylation in the context of E2 purification? As far as I understand, you are purifying with the His-tag?
Direct biotinylation system was originally developed to construct a biotin-tagged version of E2 to serve as a bait in pull-down assays (see lines 462-463). Biotinylation of E2 takes place in the secretory pathway as the biotin ligase is fused to a signal peptide. To clarify this point we describe the BirA construct in the Materials and Methods section (lines 153-155). As highlighted by the reviewer, biotinylation was not required to purify the recombinant protein.
H) Fig 4 C, the black bar in the negative control, does it refer to your detection limit or is it a contamination, or something else? Could you please clarify this?
The bar labeled - in panel C shows the quantification for mock infected cells. Reference to column label was unintentionally omitted in the original manuscript and is now included in the figure legend.
I) In Fig 5 under passage 4, do you think it is possible to adjust the grey vertical dashed lines so that they are at the same nucleotide as for the other two passages? I think this would make it easier for the reader.
Unfortunately, we were unable to adjust the dashed lines.
Reviewer 5 Report
The mechanism of Pestivirus entry is not well understood, however it is well established that the envelope glycoprotein 2 (E2) is critical for host receptor interaction. The crystal structures of BVDV E2 were determined several years ago by two independent groups. The E2 has two IgG like domains and a unique, elongated domain. Within the second IgG like domain there is a elongated beta hairpin that has been reported to be involved in receptor interaction. In this manuscript the authors have made specific point mutations to conserved amino acids within the loop and assayed it for cell binding and BVDV infection.
The major scientific question:
The cell binding assay of purified BVDV E2 onto MDBK cells is interesting and potentially important for further studies, however a control is needed since glycosylated proteins can be endocytosed in a nonspecific mechanism. Is there a way to show specificity for the interaction? For example, does the addition of a neutralizing or other E2 antibody prevent the interaction? Does a CD46 antibody inhibit binding? Lastly, is there binding to a nonpermissive cell line?
Minor remark:
Please include residues numbers and accession codes for the sequences in the alignment for Figure 1 A.
Author Response
The major scientific question:
The cell binding assay of purified BVDV E2 onto MDBK cells is interesting and potentially important for further studies, however a control is needed since glycosylated proteins can be endocytosed in a nonspecific mechanism. Is there a way to show specificity for the interaction? For example, does the addition of a neutralizing or other E2 antibody prevent the interaction? Does a CD46 antibody inhibit binding? Lastly, is there binding to a nonpermissive cell line?
Non-specific endocytosis of E2 in cell binding assays was addressed during assay optimization. We tested binding at different temperatures and using different protein concentrations to achieve sensitive detection of E2 on the cell surface. Indeed, in flow cytometry assays E2 staining is performed in non-permeabilized cells. We observed dose dependent signal intensity for both the wild type and the mutant protein and higher signal after incubation at room temperature vs. incubation on ice. In line with the concern raised by the reviewer, we included experiments showing that the wild type protein does not bind non-susceptible CRIB cells of bovine origin or HeLa cells (Figure 3D) as an approach to show specificity for the interaction. We acknowledge that addition of antibodies against E2 and more importantly antibodies against CD46 would aid both in confirming the specificity of the interaction and assessing the contribution of CD46 to E2 binding. Unfortunately, we were not able to run these experiments with the tools currently available in our laboratory because sensitive detection was only achieved with a mouse polyclonal antibody against E2, and antibodies against E2 and CD46 were also raised in mouse.
Minor remark:
Please include residues numbers and accession codes for the sequences in the alignment for Figure 1 A.
Figure 1A was modified accordingly.
Round 2
Reviewer 5 Report
The authors have addressed my concerns.
Author Response
We thank the reviewer for his comments which helped to improve our manuscript.